# Effectiveness of Warm-Up Intervention Programs to Prevent Sports Injuries among Children and Adolescents: A Systematic Review and Meta-Analysis

**DOI:** 10.3390/ijerph19106336

**Published:** 2022-05-23

**Authors:** Liyi Ding, Jianfeng Luo, Daniel M. Smith, Marcia Mackey, Haiqing Fu, Matthew Davis, Yanping Hu

**Affiliations:** 1P.E. College, Shanghai Normal University, Shanghai 200234, China; 2School of Public Health, Fudan University, Shanghai 200032, China; jfluo@shmu.edu.cn; 3Department of Physical Education and Health Education, Springfield College, 263 Alden Street, Springfield, MA 01109, USA; dsmith19@springfield.edu (D.M.S.); mdavis2@springfield.edu (M.D.); 4Department of Physical Education & Sport, Central Michigan University, Mt. Pleasant, MI 48859, USA; macke1mj@cmich.edu; 5P.E. Department, Fudan University, 130 Dong An Road, Xuhui District, Shanghai 200032, China; fuhq@fudan.edu.cn; 6The Center of Disease Control & Prevention, Putuo District, Shanghai 200333, China; huhu1121@hotmail.com

**Keywords:** warm-up intervention program, adolescents, children, sports injuries, sports activity

## Abstract

Sports participation by children and adolescents often results in injuries. Therefore, injury prevention warm-up programs are imperative for youth sports safety. The purpose of this paper was to assess the effectiveness of Warm-up Intervention Programs (WIP) on upper and lower limb sports injuries through a systematic review and meta-analysis. Searches for relevant studies were performed on PubMed, EMBASE, Web of Science, SPORTDiscus, and Cochrane databases. Studies selected met the following criteria: original data; analytic prospective design; investigated a WIP and included outcomes for injury sustained during sports participation. Two authors assessed the quality of evidence using Furlan’s criteria. Comprehensive Meta-Analysis 3.3 software was used to process and analyze the outcome indicators of the literature. Across fifteen studies, the pooled point estimated injury rate ratio (IRR) was 0.64 (95% CI = 0.54–0.75; 36% reduction) while accounting for hours of risk exposure. Publication bias assessment suggested a 6% reduction in the estimate (IRR = 0.70, 95% CI = 0.60–0.82), and the prediction interval intimated that any study estimate could still fall between 0.34 and 1.19. Subgroup analyses identified one significant moderator that existed in the subgroup of compliance (*p* < 0.01) and might be the source of heterogeneity. Compared with the control group, WIPs significantly reduced the injury rate ratio of upper and lower limb sports injuries in children and adolescents.

## 1. Introduction

Sports participation is one of the main factors resulting in injuries in children and adolescents. Therefore, injury prevention warm-up programs are imperative for youth sports safety. According to findings from the 1998 World Health Organization, a cross-national study on the health behavior of school-age children found that 21.8% of the injuries in children at the ages of 11, 13, and 15 years were injuries that occurred during sports and playground activities [1]. Another study found that most injuries were experienced by male students aged 10–14 years, with falls and sports injuries being the most common injury mechanisms [2]. Therefore, intervention programs aimed at preventing school sports injuries in children and adolescents are very important to reduce the personal and social costs associated with treatment and rehabilitation. Similarly, they help to maintain the positive outcomes of exercise participation, such as obesity reduction, cardiovascular health promotion, and skill development.

Although there have been many randomized controlled trials of warm-up intervention programs (WIP) for the prevention of sports injuries among children and adolescents in recent years, there has been some inconsistency in the findings. Some studies concluded that the WIPs were effective for the prevention of sports injuries. For example, Owoeye et al. [3] showed that neuromuscular exercises significantly reduced the risk of ankle sprain in juvenile football and basketball players. Hornbeck et al. [4] found that neuromuscular programs can reduce the knee joint injury of adolescent female football players, and the injury rate of the anterior cruciate ligament of the knee joint can be reduced by 64% (rate ratio 0.36, 95% CI: 0.15–0.85). Beaudouin et al. [5] found evidence that an intervention program for children 11 years or older lasting 15–20 min could significantly prevent serious sports injury of children football players (hazard ratio 0.42, 95% CI 0.24 to 0.72). However, some studies [6,7] have shown the opposite. For instance, Steffen et al. [6] carried out an 8-month warm-up intervention including core stability, lower limb strength, neuromuscular control, and other exercises for young female football players. The results showed there was no significant difference in the overall injury rate (RR = 1.0, 95% CI 0.8–1.2; *p* = 0.94) between the intervention group (M = 3.6 injuries/1000 h) and the control group (M = 3.7 injuries/1000 h). In addition, Zakaria et al. [7] also found there was no significant difference between dynamic stretching and dynamic plus static stretching in the prevention of lower limb, core and back injuries in high school male football players (*p* = 0.33). Therefore, it is necessary to further analyze the research findings through meta-analysis.

The purpose of this study was to determine the effectiveness of WIPs reported in the literature. Through meta-analysis, comparisons were made between WIPs and warm-up as usual (e.g., only running and stretching). Specifically, the research focused on three kinds of WIPs, which included comprehensive, neuromuscular, and balance warm-ups. In addition, the moderating effects of participants (e.g., age, gender, sport level), background (e.g., settings, sport type), and WIP variables and characteristics (e.g., content, compliance) on injury risk reduction were assessed. The results of the current study provide some insights to guide the development and implementation of effective WIPs in the prevention of sports injuries for children and adolescents.

## 2. Materials and Methods

### 2.1. Definition of Terms

In accordance with the definition by Emery et al. [8], sports injury was defined as any injuries sustained through a sport or recreational activity in or outside of school that resulted in missed time from activity participation (unable to return to the same session or prevented future activity participation) or required medical attention. Therefore, any types of injuries that met this definition were included.

WIPs in the study were defined as a series of physical exercises including strength, balance, aerobics, stretching, etc. performed at the beginning of each session in order to increase body temperature [9] and prevent sports injuries [10], such as a comprehensive program, neuromuscular program, or balance program.

### 2.2. Search Strategy

A systematic literature search was conducted in December 2021, and a systematic review and meta-analysis were reported according to the Preferred Reporting Items for Systematic Reviews and Meta-analyses (PRISMA). This study was registered at the International Prospective Register of Systematic Reviews (Number CRD42020163514). The following bibliographic databases were used: PubMed, EMBASE, Web of Science, SPORT Discus, and Cochrane Central Register of controlled trials. Concurrently, the bibliography of relevant articles was manually searched to find other potential references. Keyword search terms (including derivatives) are as follows: child, adolescents, intervention/prevention, athletic injuries, randomized controlled trial. Detailed keywords, variants used, and the number of articles retrieved from the above five databases are included in the Appendix B.

### 2.3. Selection of Studies

Study inclusion criteria were: (i) contained original data (full-text paper published); (ii) investigated an outcome of sports injuries including school sport and non-school sport settings; (iii) evaluated an injury prevention intervention of warm-up including neuromuscular program, comprehensive program, or balance program; (iv) included sport participants 18 years of age or younger; (v) analytical study design (including RCT, Cluster RCT); (vi) peer-reviewed. Exclusion criteria were: (i) non-randomized controlled trials, case reports, published abstracts, conference proceedings, or reviews; (ii) participants were university students (above 18 years old); (iii) lack of a control group; (iv) the outcome of school sport or non-school sports injuries was concussion only.

### 2.4. Data Extraction

Two reviewers (HF and MD) used a specially designed standardized data extraction table to extract data independently and then compared the extracted data to ensure consistency. All inconsistencies between the two tables were resolved through a discussion between the two reviewers. After initially discussing the inconsistency between two separate reviewers, any differences in the data extraction process were resolved by a third party (LD). Data for each study were extracted from the full text. If there was insufficient data, the author was contacted by email.

### 2.5. Quality Evaluation of Selected Studies

Two reviewers (HF and MD) independently scored the methodological quality of the included trials using 12 quality criteria adapted from Furlan et al. [11]. The criteria are randomization, concealed allocation, blinding of participants, blinding of care providers, blinding of outcome assessors, drop-out rate, analysis according to the allocated group, reporting without selective outcome, baseline similarity of the groups, co-interventions, compliance, and timing of outcome assessment.

### 2.6. Data Analysis

#### 2.6.1. Meta-Analysis

All data were calculated and analyzed by Comprehensive Meta-Analysis 3.3 (Biostat, Inc., 2014). The injury rate ratio (IRR) and its 95% confidence interval were used as the effect measure for each study. The formula for calculating is: IRR = (number of injuries in intervention group/hours of total exposure)/(number of injuries in control group/hours of total exposure). The statistical method was the DerSimonian and Laird (D-L) method. The analysis model was a random effect model. The overall IRR point estimation and 95% confidence interval indicated the pooled overall effect, while the Z and *p* values tested the null hypothesis that the injury rate ratio estimate was no more effective than the control.

#### 2.6.2. Heterogeneity

Statistical heterogeneity (I^2^) and overall effect test were calculated, and the threshold value of 0.05 was used for statistical significance. Q Statistics (with df and *p* values) were examined to provide a test of the null hypothesis that all studies shared a common effect size. If all studies shared a similar effect, the Q value would be approximately equal to the degrees of freedom. The I^2^ statistic identified the proportion of the observed variance that reflected the differences in true effect sizes as opposed to sampling error. In the subgroup analysis, a medium-high value (>0.50) was used as the possible explanation of the source of heterogeneity. T^2^ provided the estimate of the between-study variance in true effects, and T provided the estimate of the between-study standard deviation in true effects.

#### 2.6.3. Subgroup Analysis

In order to examine the sources of heterogeneity and identify potential moderators of injury rate ratios, several exploratory subgroup analyses were conducted using mixed-effect analysis and random effect models. This assumed a common among-study variance component across subgroups. Due to the limited number of studies available (i.e., lack of sufficient power), meta-regression options were excluded. Subgroup analysis was considered when I^2^ > 50% or Q statistic test *p* < 0.05. Identified and verifiable factors led to 8 independent subgroup analyses (e.g., age, gender, settings, level, WIP type, injury site, participant compliance, study quality). The Q, df, and *p* values between groups determined whether a specific subgroup injury rate ratio and (95% CIs) were more associated with injury reduction. Evidence of the dispersion in true effects among subgroups was also scrutinized.

#### 2.6.4. Publication Bias

Standard funnel plots were used to detect evidence of publication bias. The Egger test [12] was then applied to confirm the asymmetry and the “trim and fill” procedures [13] were performed if the overall IRR estimate needed to be adjusted for missing studies.

#### 2.6.5. Sensitivity Analysis

The sensitivity analysis of this study mainly adopted two methods: (1) when choosing different statistical models, test the difference between the point estimation and the interval estimation of the pooled effect size; (2) test the change in results after deleting the maximum and minimum value of the IRR.

## 3. Results

A total of 2132 articles were generated by restricting non-English and non-peer-reviewed journals. First, the duplicates were deleted among the retrieved articles and then evaluated according to the title and abstract. A total of 431 articles were obtained. Second, studies were excluded if they did not meet the inclusion criteria. Third, 52 full-text journal articles remaining were assessed. Finally, after eliminating studies of non-warmup intervention, secondary analysis of pooled data from multiple studies, and studies lacking statistics on sports injury, the final 15 articles were used for meta-analysis. The literature search process is shown in Figure 1.

### 3.1. Risk of Bias

The risk of bias was assessed by the score of the quality assessment form. All items were scored as follows: + = yes (1 point), − = no (0), ? = unable to determine (0). The scores for each study were summarized in Table 1. If the score of the study is greater than 50% of the highest score, it is considered “high quality” [14]. If, due to insufficient information in the report, reviewers were unable to determine the score for a particular criterion, they would try to contact the author. If ambiguities or disagreements persisted, the third-party coauthor (LD) would be consulted. All of the 15 studies scored at least 6 of 12 on the scale, so they can all be considered “high quality”. The absence of blind intervention for intervention group participants and caregivers (who perform the intervention) was a common methodological defect.

### 3.2. Research Features

In 15 identified studies, 3 studies used male participants, 3 studies used only female participants and 9 studies used mixed-gender samples. Participants were primary and secondary school students aged 7–18 years. From the perspective of nationality, participants were mainly from the United States, Canada, Norway, Italy, the Netherlands, Finland, Switzerland, Germany, and the Czech Republic. In terms of sports types, 11 studies were single sports, such as soccer, basketball, floorball, and handball, while 4 studies were multi-sports, such as physical education and different combinations of sports (Table 2).

Further details on the characteristics of each study were presented in Table 3. In terms of the contents of WIP in the identified studies, three types of intervention programs could be determined according to their contents: comprehensive intervention program (9 studies), neuromuscular program (5 studies), and balance program (1 study). The average duration of sports injury prevention intervention was 7.4 months.

*Comprehensive program*. The comprehensive programs typically were FIFA 11, FIFA 11+, or related programs, which included aerobic exercise, strength, jumping and balance, and self-protection when falling. They usually lasted about 15–20 min [15,16].

*Neuromuscular program*. The neuromuscular program was mainly in the form of warm-up activities lasting approximately 15 min. The contents of WIPs mainly included strength, flexibility, balance, aerobic, flexibility, core strength, and muscle strengthening [8,17,18,19]. In addition, one study adopted a combination of both in and out-of-class activities [20].

*Balance program*. The balance program mainly included aerobic exercise, dynamic stretching, strength exercises, single leg balance (eyes open/closed), stance balance on the wobble board, etc. The intervention time was about 10–15 min, in addition to 20 min of wobble board exercise at home [21].

### 3.3. Meta Analysis: Injury Rate Ratio

Figure 2 provides a summary of the data entered and IRRs for each study, as well as the pooled estimate. Based on identified studies sampled from a range of possible studies, the pooled data reflected 21,576 child and adolescent participants (age range, 7–18 years) covering 110.5 months of intervention with 3910 injuries. Compared with the control group, the overall IRR was 0.64 (95% CI = 0.54–0.75), indicating that the implementation of WIP reduced sports injury by 36%. The prediction interval (0.34–1.19) indicates that the real effect of any study may be in the range of these values. The null hypothesis was rejected (z = −5.492, *p* < 0.001), which led us to conclude that the WIP significantly reduced the rate ratio. The Q value of 78.74 (df = 14, *p* < 0.001) indicated variability in the real effect size across studies. I^2^ of 82.22 suggested there was a large heterogeneity among the studies, so subgroup analysis was necessary. Meanwhile, T^2^ and T were 0.076 and 0.276 (logarithmic units), respectively.

### 3.4. Publication Bias

Examination of the funnel plot indicated no deviations (see Figure 3). Based on the studies identified, asymmetry was apparent, as smaller studies typically had higher-than-average effect sizes, with low effect sizes absent. Moreover, the Egger test [12] confirmed the asymmetry (intercept = −4.26, SE = 1.38, *p* = 0.009). The ‘‘trim and fill’’ procedure of Duval and Tweedie [13] provided an adjusted overall IRR of 0.70 (95% CI = 0.59–0.82; *n* = 3 imputed studies), indicating that a minor adjustment to the overall point estimate was warranted. The adjusted point estimate remained fairly close to the original estimate.

### 3.5. Subgroup Analysis

Table 4 displays summaries from the mixed-effects analysis, as well as the results of applying the random-effects model based on the eight moderating factors entered. Although all of the subgroups were mainly associated with IRR reduction, Q and P values for between-subgroups comparisons indicated only one discernible difference in compliance (*p* = 0.01). It indicated the source of heterogeneity among the studies identified. In addition, two subgroups in the meta-analysis showed more than 10% differences in IRR point estimates, for example: the compliance of >70% (IRR = 0.56, 95% CI 0.48–0.67 vs. <70% = 0.81, 95% CI 0.65–1.01) and gender (IRR: male = 0.47, 95% CI 0.36–0.62 vs female = 0.68, 95% CI 0.44–1.04 and mixed = 0.67, 95% CI 0.55–0.81).

### 3.6. Sensitivity Analysis

Sensitivity analysis was performed for all included studies. First, the fixed-effect model and the random effect model were compared. The point estimate of the IRR of the fixed-effect model was 0.72 (95% CI 0.67–0.76), and the point estimate of the IRR of the random effect model was 0.64 (95% CI 0.54–0.75), so the difference between the two models was not significant. Second, after deleting the maximum [22] and minimum [19] IRRs, the pooled overall odds ratio was estimated to be 0.64 (95% CI 0.56–0.74), and the results remained similar.

## 4. Discussion

The aim of the present study was to assess the effectiveness of WIP on the injury reduction rate of children and adolescents in both school-based and non-school-based sports. When systematic search and meta-analysis procedures accounting for exposure hours were applied, the pooled estimate of the injury risk reduction was about 36% (pooled IRR = 0.64, 95% CI = 0.54–0.75), or 30% if adjusted for bias. This finding represents a statistically significant and clinically meaningful reduction in injury rates. When consistent with the relevant meta-analysis, the estimated reduction rates were similar for people with different age and skill levels, or for those with exercise interventions. For example, in the case of 21 adolescent studies, Rossler et al. [23] found a significant reduction in overall injury (RR = 0.54, 95% CI = 0.45–0.67, *p* < 0.001). Lauersen et al. [24] showed that exercise intervention reduced the risk of acute injury by 35.3% (RR = 0.65, 95% CI = 0.50–0.84, *p* < 0.001), while Hubscher et al. [25] found that multiple intervention exercises could effectively reduce the risk of lower limb injury (RR = 0.61, 95% CI = 0.49–0.77, *p* < 0.01), and balance exercises alone could significantly reduce the risk of ankle sprain (RR = 0.64, 95% CI = 0.46–0.90, *p* < 0.01).

Warm-up is a series of physical exercises performed before a more vigorous exercise. Warm-up can be either passive or active [26]. Active warm-up can further be classified as either a general warm-up or a specific warm-up. A general warm-up includes jogging, stretching, calisthenics, and some resistance exercise. A specific warm-up includes specific stretches and movements that will be used in the sport. A passive warm-up is one in which muscle temperature or core body temperature is increased by external means, which can include, for example, hot showers, saunas, or heating pads [26]. Therefore, all of the studies in the meta-analysis consisted of active warm-ups.

Components of the WIPs in the 15 studies were strength, aerobics, balance, stretching exercises, self-protection skills, and specific sports movements. Among them, almost all the studies involved strength exercises in their WIPs except for one study [21]. WIPs primarily focused on the lower limbs and core strength, for example, plank, side plank, Nordic hamstring lower, squat, static lunges, walking lunges, heel raises, etc.; 11 studies involved aerobic exercise [8,15,19,20,21,22,27,28,29,30,31]. All the WIPs adopted in the study were in the form of warm-up, so they usually included jogging, forward running with knee lifts, forward running with skipping, sideways shuffles, various types of jumping exercises, etc. There were 4 studies involving balance exercise [8,15,19,21], the purpose of which was to improve proprioceptive ability and avoid the risk of falling, which included single leg balance, two feet balance on the wobble board, single leg balance on the balance pad, while completing various functional actions such as dribbling, catching, and kicking under the condition of single leg balance. Five studies adopted stretching exercises [20,21,22,28,29], which included static stretch and dynamic stretch components, for example, lunges and walking lunges, etc. There were **five** studies that involved specific sports exercises. For example, in the FIFA 11+ programs, some movements of running and cutting in the warm-up were usually employed in the sport of soccer, which can prepare participants for the training in advance. Finally, in order to prevent accidental sports injuries, **three** studies involved self-protection skills in the warm-up [16,29,31].

In the age subgroup analysis, it was found that the intervention effect of WIPs on middle school students was better than that of primary school students (IRR = 0.63 vs. 0.68), which may be related to the younger age of primary school students and less exercise experience. Children more commonly lacked the awareness of self-protection during their exercises, so they were more likely to experience unintentional injuries in sports [32].

In the subgroup analysis of intervention type, it was found that the intervention effect of neuromuscular/balance exercise was better than comprehensive programs (IRR = 0.59 vs. 0.66). In the intervention study of neuromuscular exercises, Richmond et al. [19] adopted the WIP including aerobic exercise, core/lower limb strength exercise and balance exercise for 10 min, and achieved very good efficacy (IRR = 0.30, 95% CI 0.19–0.47). Among the subgroups of the comprehensive intervention program, Longo et al. [33] also achieved the best intervention effect in this subgroup by using the FIFA 11+ comprehensive program (IRR = 0.44, 95% CI 0.22–0.89). Their comprehensive intervention program mainly included three parts: (i) running exercises at slow speed combined with active stretching and controlled contact with a partner; (ii) a different set of exercises, including strength, balance, jumping exercises, and Nordic hamstring exercises; (iii) speed running combined with basketball-specific movements with sudden changes in direction. Through the comparison of the studies among the two subgroups, it was found that Richmond [19] found the best effect from their neuromuscular exercises because of the components in their intervention program aimed to prevent a sports injury: aerobic exercise, neuromuscular strength exercises, and balance exercises. It consisted of 10 min of continuous aerobic exercise and 5 min of core/lower limb strength, as well as balance exercises. In addition, the compliance rates of intervention schools and control schools were higher, reaching 84% and 95%, respectively. High compliance helps to ensure the effects of the intervention trial. Some studies have shown that the level of compliance would directly affect the final intervention effect of the experiment [6,34,35]. In addition, by increasing the exercise intensity to 75% of the maximum heart rate, the intervention program also achieved the effect of improving physical fitness and controlling weight.

In the subgroup analysis of settings, it was found that the intervention effect of non-school-based programs was better than that of school-based programs (IRR = 0.61 vs. 0.69). Because all the intervention studies were in the form of warm-ups and the warm-up time was generally about 15 min, the intervention program may not include all the related exercises. In addition, the intervention of a single sport (basketball, volleyball, football, or floorball) was simpler than the multi-sports (physical education), which perhaps made it easier to achieve the desired effect. The research subjects of non-school-based programs were usually high-level players competing in clubs. They had a rich experience in sports and gained a specific warm-up intervention designed for their sport, so it was common to have a better intervention effect in the non-school-based program group.

## 5. Conclusions

When synthesized across 15 cluster randomized controlled trials, the IRR of the warm-up intervention group was significantly reduced by 36% (pooled odds ratio = 0.64, 95% CI = 0.54–0.75) compared with the control group or the warm-up as usual group. Compared with the control group, WIPs significantly reduced the injury rate ratio of upper and lower limb sports injuries in children and adolescents.

## 6. Implication

The results of the current study provide some insights to guide the development and implementation of effective WIPs in the prevention of sports injuries for children and adolescents. The efficacy of WIP has important clinical and practical significance for physical therapists, physical education teachers, coaches, conditioners, school administrators, and social sports organizations engaged in children’s and adolescents’ sports activities. The WIP decreased the rate of sports injury (such as acute injury or overuse injury), thus reducing the subsequent personal, social and economic costs, including injury fixation, treatment, and rehabilitation [18,36]. As individual studies have shown, effective WIP contents were structured, multifaceted, and implemented frequently, stably, and consistently in the long term, which would lower the injury risk. The results also emphasized how to make practical adjustments in sports training and competition, as well as in family exercises. In general, clinical and sports practitioners can recommend and implement the WIP as part of the injury prevention strategy.

## 7. Research Strengths and Limitations

The strengths of the current study were different from the previous meta-analysis of sports injuries in adolescents in the following aspects: this study (i) focused on the WIPs in both school-based and non-school-based settings; (ii) examined the injury rate of specific anatomical sites, i.e., the upper and lower limbs, rather than the overall injury rate; (iii) focused on a subset of the population (i.e., children) who were more vulnerable to sports injury; (iv) incorporated the most recent studies that were from peer-reviewed journals with high impact factors.

No significant invalid results were found in all subgroup analyses, indicating that interventions in each subgroup were effective and were not limited by sample data from individual studies (e.g., sometimes <5 studies per subgroup). Both the moderating factors examined and those that could not be examined (such as exposure time, maturity, etc.) could still moderate the effect of the WIPs on IRRs. These factors may be investigated when additional data are available.

Another limitation of the study was that some upper and lower limb research results also included head and/or trunk sports injuries, but it did not influence the results because the majority of sports injuries occurred in the upper and lower limbs.

## Figures and Tables

**Figure 1 ijerph-19-06336-f001:**
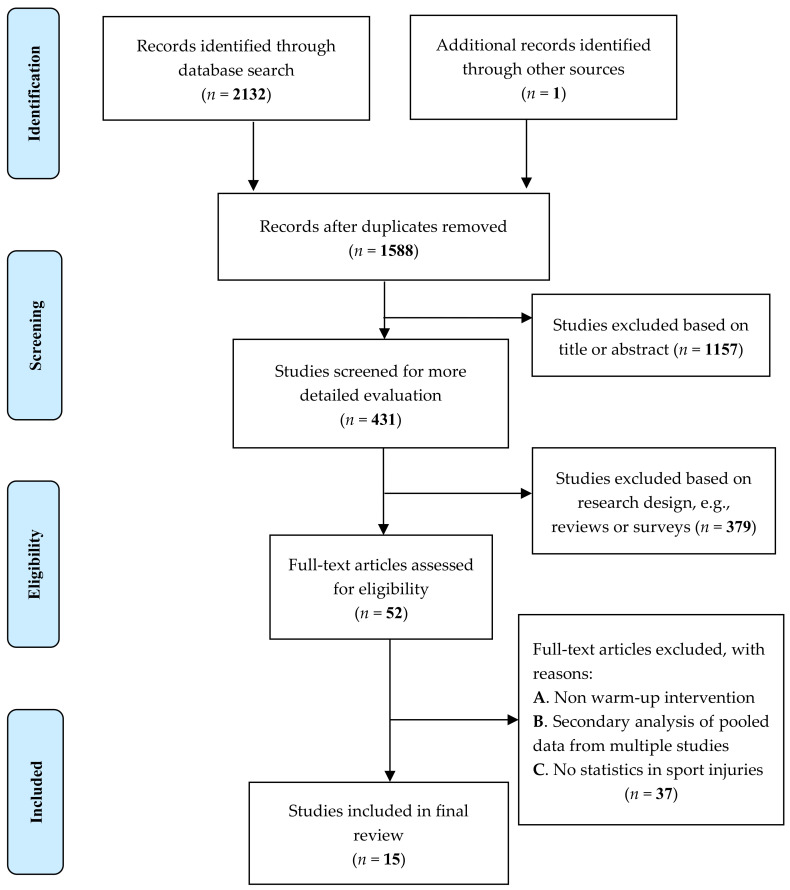
Flow diagram for screening and selection of studies according to PRISMA (Appendix A).

**Figure 2 ijerph-19-06336-f002:**
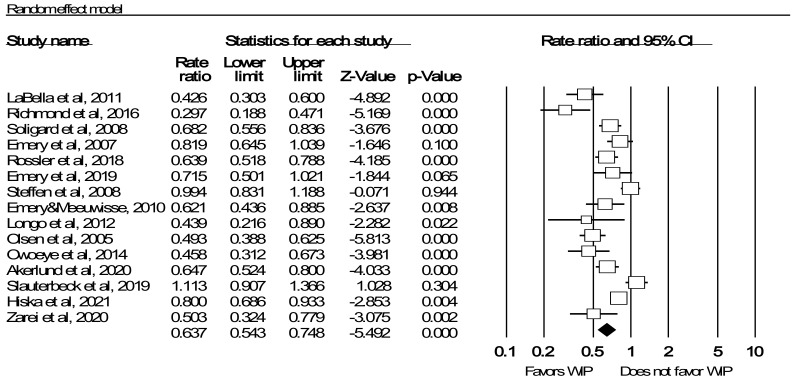
Forest plot illustrating the pooled effect of warm-up intervention program (WIP) as compared with controls on IRR.

**Figure 3 ijerph-19-06336-f003:**
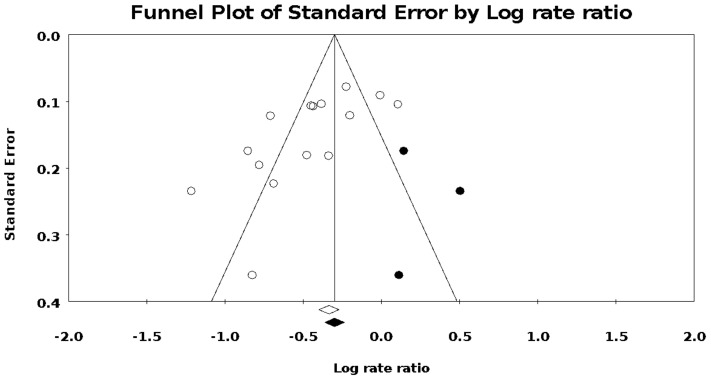
Funnel plot based on study standard error and log risk ratio in assessing publication bias.

**Table 1 ijerph-19-06336-t001:** Study Quality and Risk of Bias Assessment ^a^.

Study Name	Risk of Bias Assessment ^b^												Score	%
	1	2	3	4	5	6	7	8	9	10	11	12		
LaBella et al., 2011	+	+	−	−	−	+	+	+	?	?	+	+	7	58.3
Richmond et al., 2016	+	+	+	−	?	+	+	+	+	?	+	+	9	75.0
Soligard et al., 2008	−	+	−	−	+	+	+	+	+	?	+	+	8	66.7
Emery et al., 2007	+	?	−	−	+	+	+	+	+	?	+	+	8	66.7
Rossler et al., 2018	+	?	−	−	+	+	+	−	+	+	+	+	8	66.7
Emery et al., 2019	+	+	+	+	+	+	+	?	+	+	?	+	10	83.3
Steffen et al., 2008	−	+	−	−	−	+	+	+	+	?	+	+	7	58.3
Emery and Meeuwisse, 2010	?	+	+	+	?	+	+	+	+	?	+	+	9	75.0
Longo et al., 2012	+	+	−	−	+	+	+	+	−	+	+	+	9	75.0
Olsen et al., 2015	+	+	?	−	+	+	+	+	+	+	+	+	10	83.3
Owoeye et al., 2014	+	?	?	−	−	+	+	+	+		+	+	7	58.3
Akerlund et al., 2020	+	−	−	−	+	+	+	?	+	+	+	+	8	66.7
Slauterbeck et al., 2019	+	−	−	−	?	+	+	?	+	+	−	+	6	50.0
Hiska et al., 2021	+	+	+	−	+	+	+	?	+	−	+	+	9	75.0
Zarei et al., 2020	+	+	+	−	+	−	+	−	+	+	−	+	8	66.7

Note. ^a^ Maximum obtainable quality score, 12. +, yes (1 point); –, no (0 points); ?, unable to determine (0 points). ^b^ Risk of Bias Assessment: 1. randomization; 2. concealed allocation; 3. blinding of participants; 4. blinding of care providers; 5. blinding of outcome assessors; 6. drop-out rate; 7. analysis according to allocated group; 8. reporting without selective outcome; 9. baseline similarity of the groups; 10. co-interventions; 11. compliance; 12. timing of outcome assessment.

**Table 2 ijerph-19-06336-t002:** Overall Situation of Identified Studies ^a^.

Study	Country	Settings	Sex	Age	Sport	Level	Design	Intervention	Outcomes	Quality
LaBella et al., 2011	USA	School-based	F	N/A	S.B.	Amateur	Cluster-RCT	Neuromuscular	LE	7
Richmond et al., 2016	Canada	School-based	Mx	11–15	P.E.	Amateur	Cluster-RCT	Neuromuscular	LE and UE	9
Soligard et al., 2008	Norway	Non-School-based	F	13–17	S	Club	Cluster-RCT	Comprehensive	LE	8
Emery et al., 2007	Canada	School-based	Mx	12–18	B	Amateur	Cluster-RCT	Balance	LE	8
Rossler et al., 2018	Multi Countries	Non-School-based	Mx	7–12	S	Club	Cluster-RCT	Comprehensive	LE and UE	8
Emery et al., 2019	Canada	School-based	Mx	11–16	P.E.	Amateur	Cluster-RCT	Neuromuscular	LE and UE	10
Steffen et al., 2008	Norway	Non-School-based	F	16–18	S	Club	Cluster-RCT	Comprehensive	LE and UE	7
Emery and Meeuwisse, 2010	Canada	Non-School-based	Mx	13–18	S	Club	Cluster-RCT	Neuromuscular	LE	6
Longo et al., 2012	Italy	Non-School-based	M	13.5 ± 2.3	B	Club	Cluster-RCT	Comprehensive	LE	9
Olsen et al., 2005	Norway	Non-School-based	Mx	15–17	H	Club	Cluster-RCT	Comprehensive	LE	10
Owoeye et al., 2014	Nigeria	Non-School-based	M	14–19	S	Club	Cluster-RCT	Comprehensive	LE and UE	7
Akerlund et al., 2020	Sweden	Non-school-based	Mx	12–17	F	Amateur	Cluster-RCT	Comprehensive	LE and UE	8
Slauterbeck et al., 2019	USA	School-based	Mx	N/A	Ms	Amateur	Cluster-RCT	Comprehensive	LE	6
Hiska et al., 2021	Finland	Non-school-based	Mx	9–14	S	Club	Cluster-RCT	Neuromuscular	LE	9
Zarei et al., 2020	Iran	Non-school-based	M	7–14	S	Club	Cluster-RCT	Comprehensive	LE	8

Note. ^a^ Sport: S—Soccer, B—Basketball, V—Volleyball, P.E.—P.E. class, H—handball; F—Floorball; Ms—Multi Sports; Sex: M—Male, F—Female, Mx—Mixed; Outcomes: LE—Lower Extremity, UE—Upper Extremity.

**Table 3 ijerph-19-06336-t003:** Characteristics and Content of Identified Studies ^a^.

Study	Sex	Level	Sport	Type of WIP		Intervention Program	Session	Duration	Compliance
				Intervention Group	Control Group				
LaBella et al., 2011	F	Amateur	S.B.	NMT warm-up in progressive strengthening, plyometric, balance, and agility exercises 20 min	As usual warm-up	Neuromuscular	3/wk appr.	8 mon	80.4%
Richmond et al., 2016	Mx	Amateur	P.E.	Warm-up in NMT Ae sessions 10 min + Exercises for core/lower extremity strength and balance 5min	Warm-up in low-intensity jogging 10 min + static and Dy stretch 5 min	Neuromuscular	2–3/wk	3 mon	≥84%
Soligard et al., 2008	F	Club	S	Slow speed running + exercises for strength, balance and jumping + speed running with cutting movements 20 min	As usual	Comprehensive	≥2/wk	8 mon	77%
Emery et al., 2007	Mx	Amateur	B	Ae, St, Dy stretches 10 min + Sports-specific balance warm-up 5 min + home wobble board exercise 20 min	Ae, St, Dy stretches 10 min	Balance	5/wk appr.	12 mon	73.3%
Rossler et al., 2018	Mx	Club	S	3 exercises for unilateral, Dy stability of the lower extremities +3 exercises for body and trunk strength/stability +1 exercise for falling skills 15–20 min	As usual	Comprehensive	≥2/wk	8 mon	N/A
Emery et al., 2019	Mx	Amateur	P.E.	NMT warm- up including Ae, agility, Str and balance exercises 10–15 min	Warm- up including Ae, static and Dy stretch	Neuromuscular	≥2/wk	3 mon	77.7%
Steffen et al., 2008	F	Amateur	S	FIFA 11 20 min	As usual	Comprehensive	1/wk	8 mon	52%
Emery and Meeuwisse, 2010	Mx	Club	S	Ae, St, Dy stretches 5 min+ Strength, agility and balance 10 min + 1 home wobbleboard exercise 15 min	Ae, St, Dy stretches 15 min	Neuromuscular	3/wk	5 mon	81.25
Longo et al., 2012	M	Elite	B	FIFA 11 + 20 min	As usual	Comprehensive	3/wk	9 mon	100%
Olsen et al., 2005	Mx	Club	H	running, cutting, and landing technique as well as neuromuscular control, balance, and strength. 15–20 min	As usual	Comprehensive	1/wk	8 mon	87%
Owoeye et al., 2014	M	Club	S	FIFA 11+ 20 min	As usual	Comprehensive	2/wk	6 mon	60%
Akerlund et al., 2020	Mx	Amateur	F	Swedish Knee Control program 10–15 min + a Standardized running 5 min	As usual	Comprehensive	1.45/wk avg	6.5 mon	84%
Slauterbeck et al., 2019	Mx	Amateur	Ms	FIFA 11 + 15–20 min	As usual warmup	Comprehensive	1–2/wk	12 mon	32%
Hiska et al., 2021	Mx	Club	S	Ae, squat jump, side plank, single leg balance, walking lunges, single leg jumps, speed running 20 min	As usual warmup	Neuromuscular	2–3/wk	5 mon	63%
Zarei et al., 2020	M	Club	S	Unilateral, dynamic stability of the lower extremity, trunk strength and stability, falling technique 20 min	Aerobic, dynamic stretching and football-specific movements	Comprehensive	2/wk	9 mon	67%

Note. ^a^ Sport: S—Soccer; B—Basketball; V—Volleyball; P.E.—P.E. class; H—handball; F—Floorball; Ms—Multi sports; Sex: F—Female; M—Mixed; WIP—Warm-up Intervention Program; Str—Strength, Coord—Coordination, Flx—Flexibility; Ae—Aerobics; St—Static; Dy—Dynamic.

**Table 4 ijerph-19-06336-t004:** Subgroup Analyses According to Identified Moderating Factors ^a^.

Moderator: Subgroup (No.of Studies) ^b^	Mixed-Effects Analysis Between-Subgroup Comparison						Subgroup Heterogeneity		
	Point Estimate	95% CI	*p*-Value	Possible IRR Reduction, %	Q Value	*p* Value	Q Value ^c^	*p* Value ^d^	Subgroup I^2^
**1. Age**									
-Middle school (**12**)	0.63	0.51–0.77	0.000	37			78.73	0.00	84.95
-Primary school (**3**)	0.68	0.54–0.85	0.001	32	0.23	0.64	0.01	0.92	64.51
**2. Sex**									
-Male (**3**)	0.47	0.36–0.62	0.000	53					0.00
-Female (**3**)	0.68	0.44–1.04	0.072	32			68.03	0.00	90.35
-Mixed (**9**)	0.67	0.55–0.81	0.000	33	4.56	0.10	10.71	0.01	83.04
**3. Settings**									
-School-based (**6**)	0.69	0.49–0.95	0.025	31			64.46	0.00	89.19
-Non School-based (**9**)	0.61	0.54–0.70	0.000	39	0.36	0.55	14.29	0.00	56.08
**4. Level**									
-Club (**8**)	0.60	0.52–0.71	0.000	40			69.01	0.000	61.57
-Amateur (**7**)	0.68	0.52–0.91	0.008	32	0.55	0.46	9.73	0.002	88.19
**5. WIP Type**									
-Balance/Neuromuscular (**5**)	0.59	0.43–0.82	0.001	41			78.60	0.000	84.62
-Comprehensive (**10**)	0.66	0.54–0.80	0.000	34	0.28	0.60	0.15	0.702	82.89
**6. Injury Location**									
-Upper and Lower EXT (**6**)	0.61	0.46–0.81	0.001	39			78.57	0.000	85.33
-Lower EXT (**9**)	0.65	0.53–0.81	0.000	35	0.15	0.70	0.18	0.676	82.02
**7. Compliance**									
-<70% (**5**)	0.81	0.65–1.01	0.064	19			50.40	0.000	82.48
->70% (**10**)	0.56	0.48–0.67	0.000	44	6.24	**0.01**	28.34	0.000	67.35
**8. Study quality**									
-<60% (**4**)	0.70	0.45–1.08	0.110	30			64.95	0.000	91.47
->60% (**11**)	0.62	0.54–0.72	0.000	38	0.26	0.61	13.79	0.000	66.44

Note. ^a^ Q value, dispersion of studies about the point estimate overall or within a subgroup. I^2^, heterogeneity within a subgroup. RIP, Risk intervention program; IRR, injury rate ratio, EXT, extremities. ^b^ Random effects model. ^c^ The top value per moderator indicates Q value within subgroup heterogeneity; the lower Q value indicates between subgroup heterogeneity. ^d^. The top value per moderator indicates *p* value within subgroup heterogeneity; the lower *p* value indicates between subgroup heterogeneity.

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
