# Peer review of "Effectiveness of Warm-Up Intervention Programs to Prevent Sports Injuries among Children and Adolescents: A Systematic Review and Meta-Analysis"

_ijerph, 2022, doi:10.3390/ijerph19106336_

Round 1

Reviewer 1 Report

Throughout the introduction I detected plenty of formal errors of the king: 1) misplacement of reference number within the sentence, 2) Grammatical imprecisions (I am not an English expert, but I strongly recommend a thorough revision by an English style specialist). In comparison with other sections (much more accurate) of the manuscript, it seems that the writing of the introduction was done by a completely different person, with no revision by the other authors.

  • Line 48: “Some studies concluded the warm-up …”. I suggest: “Some studies concluded that he warm-up …”
  • Line 50: “Owoeye et al. showed that neuromuscular …” I suggest: “Owoeye et al. [Reference number here] showed …
  • Line 51: “[3] Hornbeck et al. [4]..." Clarify this mismatch (reference 3 can never began a sentence)
  • Line 54: “Beaudouin et al. [insert reference number here] found evidence …."
  • Line 57: “However, some studies [insert references -various- number here] have shown the opposite.”
  • Line 62: “Zakaria et al. [insert reference number here] also found …”
  • Line 68: “… standard practice …”. The authors should clarify what they mean by standard practice. I am requesting this clarification because “standard practice” (appears also in line 366) use to implement a sort of warm-up (appropriate or not but always a sort of warm-up). If authors clarify what they mean by appropriate “… warm-up intervention programs” (line 69-73), the opposite must be also explained.
  • Resolution of Figures 1 and 2 should be improved (at least if the actual size is respected)
  • Lines 359-360: by general definition children are not and cannot be “professional players” (adults can be). No matter how much they train, school comes first. I suggest alternative expressions like “…. Involved in elite/highly demanding competition routines ….”
  • Eliminate repetitions of messages, even textual repetitions. Of note:
    • … reduced injury rates" (line 367), "... reduced the rate of sports injury" (line 374)
    • “… reduce the personal, economic, and social costs" (line 368), "... reducing the subsequent personal, social and economic costs" (line 375)

Each one of these messages should appear only once. Or Conclusions or Implication sections but not both with practically the same words/terms.

Author Response

Hello Dear Reviewer,

Thank you so much for your suggestion regarding the improvement of our manuscript! Now I upload our revised manuscript and our answer to your questions to you for your review. We hope to get your feedback about our revisons. 

Best regards,

Liyi DING (Leo)

Reviewer 2 Report

General comments:

This is a systematic review, with meta-analysis, on the effectiveness of warm-up intervention programs (WIP) in the prevention of sports injuries in children and adolescent. It seems a well conducted SR and MA, however, I have some recommendations to improve the quality of the reporting. Most critical:

  • You are already reporting results in the Methods section when you are describing your study selection strategy, including eligibility criteria. As a general rule, leave (all) the findings to the Results section as recommended by the PRISMA initiative;
  • You use interchangeably “resistance training” and “resistance exercise” throughout the manuscript. From a perspective of exercise prescription, it seems awkward that a warm-up program includes a training component as warm-up should be a preparation for the training itself. Also, report the intensity/repetitions of the exercises and how were they monitored throughout the programs. For example, you report as part of the WIP, “light aerobic exercise”. How was light defined/monitored in the primary studies? Be analytical of the primary studies. Remember that if clinicians, coaches, trainers, athletes, etc. want to use your findings, it must be possible to replicate findings otherwise it’s not of any practical application.
  • Partially related to my previous point, what was the definition of WIP? You report (good) the definition of injury but what was considered a WIP was not. One study you have selected use partially a home-based intervention (wobble board exercises) which is very far from being considered a traditional warm-up component for sports and exercise training, even if it may fall in an exercise-based preventive program. Should it be included or excluded in the SR? In addition, the classification of a WIP as “neuromuscular”, “comprehensive” or “balance” is cloudy. When I see your description of each type of WIP they all look very similar, even if a predominantly “neuromuscular” or “balance” exercise component is added to the program. What makes them so different among each other to use a classification system of exercise components needs to be clarified;
  • Please, improve the quality of the figures. They have a poor resolution to be published in high quality scientific journal.

Specific Comments/Suggestions:

Title:

OK

Keywords:

You may want to diversify your key words as all of them are already contained in the Title. It may increase the chance of your article appear when searching in databases.

Abstract:

The prediction interval does not support such categorical conclusion. Please, amend.

Introduction:

OK

Methods:

  • What about the definition of WIP? What differentiate them from injury prevention programs?

Line 86:  PRISMA is just about the reporting of a SR not its conducting.

Lines 95 –106: You are reporting Results in the Methods section. If you are truly following PRISMA guidelines then they should appear in the Results section. Please amend.

  • The authors have used an early version of the risk of bias assessment tool by the Cochrane group. Why haven’t you used an updated one, i.e., RoB 2? (https://methods.cochrane.org/bias/resources/rob-2-revised-cochrane-risk-bias-tool-randomized-trials)

Results:

  • Please report how many participants were in each study and how many participants were in total for the SR and for the MA.

Table 2: Ae, St, Dy needs to be defined

Lines 230–231: In other words, the prediction interval is telling us that WIP may have positive or non-positive preventive effects, ie, unclear effects. Isn’t it?

Line 233: “odds ratio” or “rate ratio”?

Line 255: I think something is missing in here “(P=0.01<0.05)”.

Table 4: Please check the reporting of number of studies in each subgroup. I think the number of studies by Sex and Study Quality are conflicting with the number reported in Tables 1, 2 and 3. It should be 9 studies including only male participants. And I have counted 4 studies < 60%, not 5.

Discussion:

Line 280: What do you mean by “clinically significant”?

Lines 232–233: Is this a hypothesis set by the authors or is there strong evidence of this mechanistic effect (references are missing)? Either way several SR on this topic present conflicting summaries and such categorical sentences should be avoided and/or be discussed more deeply.

Lines 319–323: This paragraph is perhaps best suited in the strengths and limitations section

Lines 388–389: the reasoning in “(iv)”, was it an eligibility criterion? I couldn’t  see it in that section.

References:

Looks OK

Author Response

Hello Dear Reviewer,

Thank you so much for your suggestion regarding the improvement of our manuscript! Now I upload our revised manuscript and our answer to your questions to you for your review. We hope to get your feedback about our revisions. 

Best regards,

Liyi DING (Leo)

Reviewer 3 Report

The manuscript presents a systematic review and meta-analysis of the effectiveness of warm-up intervention programs to prevent sports injuries among children and adolescents. The paper sounds technically good, all evidence is sustained in experimental articles with rigorous methodology described. Below, I present some points to review.  
-         I founded three different descriptions of the aim (in lines 19-21, 67-73, and 275-277). I suggest reviewing this point and choosing one of them that better characterizes the study.
-         In the introduction, the authors present an interesting framework with empirical studies. However, some previous systematic reviews were published on this theme, and these studies were not mentioned in the framework, justifying the importance of this new study. I suggest that the authors include previous systematic reviews and meta-analysis in the introduction, signalizing the gap that justifies this new study.
-         The content of lines 74-76 presents a reflection about practical implications, and I think it sounds better in section 6. 
-         Across the manuscript, we found definitions of terms and variables in different sections (i.e. in lines 78-83, 208-223, and 290-291). I suggest reorganizing it, maybe including it in the introduction, or creating a new subsection on methodology with the definition of the variables.
-         In lines 100-101, the authors described this exclusion criterion “the outcome of school sport or non-school sport injuries related concussion only”. Apparently, before in the text, they did not justify the relevance of these criteria. I suggest reviewing this.
- Figures 1 and 2 have lower quality in the PDF. I suggest reviewing it to facilitate the visualization. 
-         In subsection 2.4 the authors described the process of data extraction. Did they use a specific statistical test to check the interobserver agreement? If yes, what were the test used and the value obtained?
-         The phrase present in lines 367-369 was not explored in the discussion. I suggest authors include it there with the respective references.

Minor revisions:
-         Adjust I² in line 153.
-         Change the comma to the point after parenthesis in line 332.

Author Response

(The authors gave the same response as above.)

Round 2

Reviewer 2 Report

In general, I’m satisfied with the clarifications, including amendments, performed by the authors in this version of the manuscript. I still often find the word “training” throughout the manuscript when it is intended to designate exercise modalities during the warm-up that I believe would be more rigorous if it were replaced by “exercises” or “activities”. And the sentence “The stretching was essential to prevent muscle injuries by increasing the elasticity of muscles and smoothing muscles.” (lines 639–640) is very controversial because there is no evidence that the possible positive effects of stretching in muscle function or injury are related to changes in the mechanical properties of the muscle-tendon unit (e.g., 1, 2). I suggest rephrasing, again.

1 – Freitas SR, Mendes B, Le Sant G, Andrade RJ, Nordez A, Milanovic Z. Can chronic stretching change the muscle-tendon mechanical properties? A review. Scand J Med Sci Sports. 2018;28(3):794-806. doi:10.1111/sms.12957

2 – Chaabene H, Behm DG, Negra Y, Granacher U. Acute Effects of Static Stretching on Muscle Strength and Power: An Attempt to Clarify Previous Caveats. Front Physiol. 2019;10:1468. doi:10.3389/fphys.2019.01468

Author Response

In general, I’m satisfied with the clarifications, including amendments, performed by the authors in this version of the manuscript. I still often find the word “training” throughout the manuscript when it is intended to designate exercise modalities during the warm-up that I believe would be more rigorous if it were replaced by “exercises” or “activities”. And the sentence “The stretching was essential to prevent muscle injuries by increasing the elasticity of muscles and smoothing muscles.” (lines 639–640) is very controversial because there is no evidence that the possible positive effects of stretching in muscle function or injury are related to changes in the mechanical properties of the muscle-tendon unit (e.g., 1, 2). I suggest rephrasing, again.

1 – Freitas SR, Mendes B, Le Sant G, Andrade RJ, Nordez A, Milanovic Z. Can chronic stretching change the muscle-tendon mechanical properties? A review. Scand J Med Sci Sports. 2018;28(3):794-806. doi:10.1111/sms.12957

2 – Chaabene H, Behm DG, Negra Y, Granacher U. Acute Effects of Static Stretching on Muscle Strength and Power: An Attempt to Clarify Previous Caveats. Front Physiol. 2019;10:1468. doi:10.3389/fphys.2019.01468

Response. First, clarify three program titles as Comprehensive Program, Neuromuscular Program, and Balance Program; Second, either delete training or replace it with exercises or activities where it is applicable as suggested; Third, delete the sentence “The stretching was essential to prevent muscle injuries by increasing the elasticity of muscles and smoothing muscles.”

Reviewer 3 Report

I congratulate the authors on the final version of the manuscript.

Author Response

Thank you very much for your suggestions! :)

This manuscript is a resubmission of an earlier submission. The following is a list of the peer review reports and author responses from that submission.